# New Technological Approach for the Evaluation of Postural Control Abilities in Children with Developmental Coordination Disorder

**DOI:** 10.3390/children9070957

**Published:** 2022-06-26

**Authors:** Giada Martini, Elena Beani, Silvia Filogna, Valentina Menici, Giovanni Cioni, Roberta Battini, Giuseppina Sgandurra

**Affiliations:** 1Department of Developmental Neuroscience, IRCCS Fondazione Stella Maris, Viale del Tirreno 331, Calambrone, 56018 Pisa, Italy; giada.martini@fsm.unipi.it (G.M.); elena.beani@fsm.unipi.it (E.B.); valentina.menici@fsm.unipi.it (V.M.); giovanni.cioni@fsm.unipi.it (G.C.); giuseppina.sgandurra@fsm.unipi.it (G.S.); 2Department of Clinical and Experimental Medicine, University of Pisa, Via Roma 67, 56126 Pisa, Italy; silvia.filogna@fsm.unipi.it; 3Department of Translational Research and of New Surgical and Medical Technologies, University of Pisa, Via Risorgimento 36, 56126 Pisa, Italy

**Keywords:** developmental coordination disorder, postural control, virtual reality rehabilitation system (VRRS), quantitative assessment, daily life

## Abstract

Background: Developmental Coordination Disorder (DCD) causes difficulties in postural control which are crucial to assess due to their impact on everyday life. There is a lack of suitable tools to acquire quantitative data and deeply analyze postural control, especially during the developmental age. The aim of this study is to investigate postural control skills in children with DCD and typically developing children (TD) using the Virtual Reality Rehabilitation System (VRRS). Methods: 18 children with DCD and 30 TD children (mean age 9.12 ± 2.65 and 7.12 ± 2.77 years, respectively) were tested by using the Movement Assessment Battery for Children Second Edition (MABC-2) and a VRRS stabilometric balance platform. A *t*-test was performed to identify differences in the VRRS parameters between the two groups. Furthermore, we investigated whether a correlation exists between the VRRS data and the MABC-2. Results: Significant differences (*p* < 0.05) in mean distance and frequency of the COP are found in the two groups. These parameters also correlate with the MABC-2 total score (*p* ≤ 0.05) and balance subscales (*p* ≤ 0.05). Conclusions: This study opens a new frontier for the assessment of postural skills in children with DCD and represents a potential basis for a tailored rehabilitation program, from which their postural stability and, consequently, their everyday life will benefit.

## 1. Introduction

Developmental Coordination Disorder (DCD) is a neurodevelopmental disorder characterized by difficulties in performing fine-motor and gross-motor tasks. It is classified in the Diagnostic and Statistical Manual of Mental Disorders (DSM-5) [1] based on the following four main criteria: (1) the acquisition and execution of motor skills are significantly lower than expected for chronological age and they are manifested by clumsiness, slowness, and imprecision in activities; (2) motor difficulties interfere in activities of daily life in all of the child’s living environments (school, home, etc.); (3) symptoms of problems in motor skills occur in the early period of development; (4) the deficits cannot be explained by intellectual disability or visual impairment and they are not attributable to a neurological condition (e.g., cerebral palsy, muscular dystrophy, degenerative disorder) [1]. The prevalence of DCD in children aged 5 to 11 years is about 6% and 73–87% of children with DCD have difficulties with balance and postural control [2]; given its importance in one’s daily life, any impairment in postural control can limit children’s activity and participation [3]. Postural control is, in fact, the ability to control the body and its various segments in static and dynamic conditions [4]. Its development begins at birth and gradually improves with age. In the first two years of life, there is a rapid evolution of posture from a horizontal positions (supine, prone) to the antigravity position. However, it is around the age of 6–7 years that postural control is almost completely stabilized, although it will continue to refine during adolescence [5]. By the age of 12, children can acquire quasi-complete postural control skills [6,7]. The development of postural control takes time, as this involves different systems (vestibular, visual, and somatosensory systems) being coordinated and integrated with each other [4]. Given the importance of postural control during a child’s development, it is essential to have specific tools that allow the clinician to assess and quantify it [8]. In general, postural control can be assessed with non-instrumented and instrumented tests [8]. Non-instrumented tests mainly include functional balance tests, such as the Berg Balance Scale [9], the Timed Up and Go test [10], the Limits of Stability (LOS) [11], and the Functional Reach Test [12]. However, these tests were developed for adults and were further adopted for the assessment of the pediatric population [8]. Following the interest shown in also assessing children, an adaptation of a test for adults was designed: the Pediatric Balance Scale [13]. However, this test is not widely used in clinical practice.

It must be said that there are no specific tools for the assessment of postural control in children with DCD. This is always evaluated with developmental scales or by carrying out tests which focus on the global child’s ability, including specific assessments, such as balance subtests. The following are the tests and questionnaires which are most commonly used: the Movement Assessment Battery for Children Second Edition (MABC-2) and the Movement Checklist [14]; the Bruininks-Oseretsky Test of Motor Proficiency-2 (BOT-2) [15]; the DCD Daily [16]; and the DCD Questionnaire 2007 (DCDQ′07) [17].

Together with the cited tools, which provide mainly operator-dependent information, quantitative measurements are useful to better evaluate the subject’s performance. In this panorama, the instrumented tests make it possible to conduct kinetics, kinematics, and electrophysiological analyses. The instrumented tool most commonly used to quantify the stability in a standing position [18,19,20] is posturography, which is already used in children. Specifically, during the posturographic examinations, the amount of body oscillation is determined by recording the Centre of Pressure (COP) displacements over a period of time [5]. In general, the posturographic measurements are carried out by using force platforms, which are tools able to accurately measure the action–reaction forces between the subject and the support base. Since measuring the aforementioned forces provides an amount of information both on COP displacements and postural stability, the force platforms are the most suitable devices to assess the postural control in a quantitative way. As a result, they are considered the gold standard to refer to in obtaining reliable balance data. [5]. In addition, force platforms are expensive and heavy to transport, making their use in a non-clinical environment difficult and complicated [21]. To overcome these problems, to date, technologies provide different solutions, such as the use of virtual reality (VR) systems [22]. In this context, the Virtual Reality Rehabilitation System (VRRS) developed by Khymeia (Padova, Italy) is a medical device which allows the clinician to assess, rehabilitate or tele-rehabilitate several skills. In particular, it creates the opportunity to assess postural control and acquire quantitative data. In addition, the VRRS is designed to be portable and is therefore suitable for use outside of a clinical setting. Recently, clinicians have begun to use it among the pediatric population to carry out a more precise assessment of children’s postural control abilities [23] and to tailor tele-rehabilitation treatments [24].

Therefore, the aim of this study is to investigate and analyze postural control abilities in children with DCD compared to age-matched peers with TD, using the VRRS system.

## 2. Materials and Methods

### 2.1. Participants

The sample for our study was recruited at IRCCS Fondazione Stella Maris (Pisa, Italy) and at the inclusive roller-skating school ASD Rotellistica Camaiore (Camaiore, Italy). Consent was asked of 48 families and the trials were explained to the children while showing them the VRRS system and all of them accepted to take part in the study. The study has been approved by the Tuscany Paediatric Ethics Committee (221/2020) on 29 September 2020. The group of participants with DCD was composed of 18 children aged from 4.84 to 14.05 years (mean age ± SD: 9.12 ± 2.65 years). The group of Typically Developing (TD) children was composed of 30 children (mean age ± SD: 7.12 ± 2.77 years).

The inclusion criteria were:

For the TD group: (1) aged between 4 and 15 years old; (2) no history of developmental delay or clinically documented disorder; (3) beginners in roller-skating (no previous courses on skating).

For the DCD group: (1) aged between 4 and 15 years old; (2) documented diagnosis of DCD based on the criteria of DSM5; (3) for children with DCD that are recruited in the inclusive roller-skating school, beginners in the roller-skating (no previous courses on skating).

### 2.2. Measures/Assessment Protocol

#### 2.2.1. Movement Assessment Battery for Children-Second Edition (MABC-2)

MABC-2 [14] is a standardized, valid, and specific motor assessment battery, specifically suggested by the European Guidelines as a fundamental tool in the diagnostic process of DCD. The assessment test quantifies movement difficulties by specified motor tasks aimed at children and adolescents aged between 3 and 16 years. For each age group (3–6 years, 7–10 years, and 11–16 years) there are eight motor tasks collected in three areas: “Manual dexterity”, “Aiming and grasping”, and “Balance”. The administration time varies between 20 and 40 min. Scores are obtained in standard scores and percentiles, and they are interpreted using a traffic light system: a red light (scores below the 5° percentile), identifies a significant impairment of motor function; a yellow light (scores between the 5° and 15° percentile), indicates a risk of motor difficulty and the need to monitor over time; a green light (scores above the 15° percentile), signifies typical motor performance.

#### 2.2.2. VRRS Postural Control Evaluation with Stabilometric Balance Platform

To evaluate the postural control of the children recruited, the VRRS system (Khymeia, Italy) and its peripherals were used for the study. In particular, the VRRS system was composed of a main device (VRRS HomeKit tablet, Khymeia, Italy) and the related stabilometric balance platform connected to it via USB. The balance is a planar platform (l: 80 cm, w: 50 cm), made of four load cells which are able to capture the force applied and then compute in real-time the X- and Y-coordinates of the COP while the subject stands on the platform. Specifically, the X-Y coordinates characterize the static performances of the subject as they represent the medial-lateral (ML) and anterior-posterior (AP) displacements, respectively, and define the COP displacement. Basically, starting from the AP and ML raw data, the application is able to automatically calculate the main parameters related to the COP coordinates, such as COP distance measures, velocity, sway area, and frequencies [25]. Referring to [23], the derived parameters are considered to depict the postural control and are the most used in the posturographic field, such as (Table 1):-Mean Distance: mean value of the displacements from the center of the platform;-Root Mean Square of the mean Distance;-Total Excursion: length of the trajectory travelled by COP;-Velocity: oscillation velocity determined as the ratio of the total excursion and the duration of the test;-Frequency: rotational frequency defined as the ratio of the mean velocity and the mean distance.

All the VRRS parameters are saved locally by the main device and organized by the proper software in a final report in pdf format.

### 2.3. Setting and Data Collection

The assessments with the MABC-2 battery and the VRRS balance were carried out at the IRCCS Fondazione Stella Maris (Pisa, Italy) for some of the children of the DCD group, and at the roller-skating school Rotellistica Camaiore (Camaiore, Italy) for the rest of the DCD group and the TD children. Despite the different settings, all the children used the same VRRS balance system, and they all performed the same assessment protocol (Section 2.2).

In general, a therapist collected the demographic data of all the children, managed the VRRS system, and explained the task prior to the assessment. Specifically, subjects were required to stand on the stabilometric balance platform with open eyes and keep a static position while looking at a fixed dot on VRRS HomeKit tablet screen. Each trial lasted sixty seconds and was repeated six times; at the end of each trial, the system retrieved the mean values of all the parameters during the 60 s. The report containing a summary of the entire dataset of each test was extracted by the system and subsequently reported in an Excel file and analyzed.

During the administration of the MABC-2 assessment, the therapist collected the raw data, then each MABC-2 test was scored and the standard scores and percentiles were later inserted into an Excel file. All of the children included in the study were evaluated by the same therapist.

### 2.4. Statistical Analysis

For the statistical analysis, the mean values and standard deviation of the six VRRS tests were calculated and their relationship with the MABC-2 clinical scores was determined by using IBM^®^ SPSS^®^ Statistics software (IBM SPSS Statistics Version 26.0. Armonk, NY: IBM Corp). After evaluating the normal distribution of the dataset by using a Shapiro-Wilk’s test and performing a *t*-test to determine whether significant differences between the VRRS parameters of the groups occurred, a Pearson correlation analysis of the quantitative parameters obtained from the VRRS system and the MABC-2 clinical scores relating to the TD group and the DCD group was performed. To establish the strength and direction of the relation among the variables considered, the value and sign of the ρ index were evaluated. In addition, the statistical level of significance was set by the *p*-value (*p* < 0.05).

## 3. Results

The evaluation conducted with the MABC-2 shows that all the children in the DCD group have motor performance <16 percentile, while the TD children >16 percentile. None of the 48 children recruited were excluded from the study. Figure 1 shows the comparison between the VRRS mean parameters collected from the TD and DCD groups. As can be seen, the mean values and the standard deviation of the DCD group are slightly higher than those of their TD peers. However, there is a significant difference between the parameters related to the mean distance, the RMS, and the frequency of the COP (i.e., MD_COP_, RMS_COP_, FREQ_COP_) (*p* < 0.05). In addition, the AP component of the same parameters (i.e., MD_AP_, RMS_AP_, FREQ_AP_) also offers results which are statistically significant (*p* < 0.01), whereas a statistical difference is not found for the respective ML components, except for the FREQ_ML_ (Table 2).

Moreover, the Pearson correlation analysis carried out between the VRRS variables and MABC-2 scores reveals that almost all the technological scores correlate negatively with the clinical scores, except for the rotation frequency. Specifically, a significant negative correlation occurs among VRRS MD and RMS variables (concerning both COP and AP and ML components) and balance and total MABC-2 results, in terms of standard score (*p* < 0.01) and percentile (*p* < 0.05) (Table 3).

## 4. Discussion

The aim of this study was to investigate, using a technological tool, the postural control abilities of children with DCD compared to their age-matched peers. Some instruments, such as force and stabilometric platforms, and computerized dynamic posturography (CDP) machines provide a more accurate and precise quantitative assessment of postural control [8,26,27]. In our study, the use of VRRS technology proved to be applicable for all the children; they completed all six tests and no data was lost during the assessments. Furthermore, the VRRS system was transportable outside of a hospital environment and allowed the clinician to conduct assessments in the roller-skating school without interfering with the daily routine of the children enrolled.

From our results, it is observed that children with DCD have more oscillations of the COP than TD children, particularly in the anterior-posterior component (Figure 1, Table 1). The COP, that is, the force exerted by the feet on the force platform, represents the active control of the muscles necessary to control balance [28]. Therefore, higher COP values indicate more instability and difficulty during postural control, which have been described in DCD [29,30] particularly in the anterior-posterior component, and were confirmed by the data of the current work.

Another important result obtained in the present study is the negative correlation between the VRRS parameters and the Movement ABC-2 test (Table 2). In fact, as the COP and the values of its components (AP and ML) increase, the scores obtained in the Movement ABC-2 battery decrease. As we have previously said, high values of the COP and its components, AL and ML, indicate greater difficulty in postural control [8]. Therefore, the negative correlation found in our results indicates that children who have a lower total and lower balance scores on the MABC-2 battery also present less efficient postural control during the assessment with the VRRS stabilometric balance platform. This finding is very interesting because the experimental assessment with the VRRS correlates with a standardized test commonly used in clinical practice.

The correlation between new technological tools used for postural control analysis and a test used in clinical practice is not frequently reported. The Geuze study [29] explored a possible correlation between Movement ABC-2 and an assessment with balance platform, but with different conditions from our study. No correlation was found in that study [29], probably due to the short duration of the recording that might overestimate a child’s motor ability, particularly for less able subjects. Longer single trials, as in our protocol, could instead be challenging to maintain the static position, but might provide a more realistic estimation of the child’s motor ability [5].

Additionally, the repetition of the task could cause fatigue or a decrease in compliance, and this could influence their performance, so the current data are presented as a mean value of the six tasks. Therefore, given all these considerations, it is possible that we were able to capture the real postural control abilities and difficulties of children with DCD and hence our results provide a correlation with the Movement ABC clinical scale.

Regarding the differences that emerged in the frequency parameter (COP, AP and ML) between the DCD and TD groups, it is possible to state that, as there are no significant differences in velocity, the differences that emerged depend on the displacement of the COP. Indeed, where MD appears higher, the frequency values are lower. Children with DCD have less ability to control their oscillations during static balance and they present higher values of MD resulting in minor values of frequency. This is confirmed by the positive correlation found between frequency and MABC-2. Children with better motor performance and better balance skills assessed with the clinical scale have higher frequency values, as the instability of postural control and the oscillations during the assessment with the VRRS are lower. To our knowledge, the frequency parameter is poorly investigated in the literature of postural control analysis in children, particularly in the study of balance assessment in DCD. There are only a few studies considering frequency in the analysis of postural control [31]. However, these are not recent studies, and were conducted only on TD children. Therefore, the result that emerged from our study appears to be very interesting. However, further studies in the pediatric age and, above all, in children with pathologies are needed.

In our study, given the correlation between variables of the VRRS and the Movement ABC-2, the VRRS allows the clinician to explore motor skills and postural control, and to obtain objective data which is used in conjunction with clinical scales to complete the traditional assessment.

The assessment with this technological system allowed us to obtain quantitative data that accurately describes the different characteristics of balance control. In fact, we obtained information on the different spatial components of balance (anterior-posterior and medial-lateral directions), but also on the velocity and frequency of the different oscillations. Thanks to the information collected from the VRRS report we can investigate, in greater detail, the strategies adopted by children with DCD for maintaining posture in static bipedal balance. It is reported in the literature that motor control strategies for regulating muscle activity are less uniform in children with DCD than in typically developing children [32]. Several neuromuscular deficits in muscle activation during balance trials have been found in children with DCD [32,33,34] that may influence motor strategies in postural control. However, their motor control strategies are not analyzed in detail, including the hip and ankle strategies usually present in typically developing children for balance control [35,36,37]. In the future, these analyses could also be integrated.

Many of the results found in the literature for the assessment of balance in DCD come from dynamic balance tests, through balance perturbations. For this reason, we wanted to investigate what happened in the case of a static balance evaluation.

However, this work has some limitations which are as follows: (i) We did not analyze postural control abilities by performing tests under different sensory conditions, such as with their eyes closed, as often explored in the DCD literature [29,38,39]. Even if this point could be considered a minor limitation, perceptual skills, visual-spatial processing, and visual-kinesthetic integration have been shown to be impaired in children with DCD and they are important skills and prerequisites in maintaining balance and postural control [40]. Therefore, we eliminated all sensory conditions, proposing static bipedal balance tests with open eyes. This choice was made because we wanted to analyze the postural control abilities in children with DCD compared to a TD group in the most favorable condition and without including variables that could involve other systems; (ii) We did not consider or analyze the possible co-morbidities of the DCD group (i.e., Attention Deficit Hyperactivity Disorder (ADHD), mild intellectual disability, or mild Autism spectrum disorders) and their influence on the balance tests, since some studies [41,42] show their role; (iii) the study samples included a small number of participants with different age mean; (iv) We did not consider the possible previous experience of participants in using commercial devices to train the postural control skills and the capability of this to impact their performances.

Despite these limitations, the strong point of this study is the promising results regarding the use of technologies in the pediatric age. In fact, in this study, the VRRS proves to be a useful instrument for the evaluation of postural control. The VRRS also presents a rich and variable exercise library, which can be used to provide customized rehabilitation treatment. In particular, it is possible to perform exercises in which COP displacements are only performed in one direction, such as anterior-posterior or medial-lateral, or in two directions together (anterior-posterior and medial-lateral), but also omnidirectional. In addition, all these exercises are included in VR, with attractive and motivating graphics for children [43]. Therefore, as a technological tool, the VRRS system could be a new frontier for personalized, more motivating, and exciting rehabilitation programmes for these children, thanks to the different information provided by the quantitative data, such as the analysis of the different balance components. Recent studies have used the VRRS with excellent results, not only for adult motor and cognitive rehabilitation [44,45], but also in children [24,46]. It is shown that VR and the use of technology motivate these children during treatments and improve their balance skills, as reported in several studies [47,48,49,50]. However, in the existent literature, training is provided by means of commercial devices (such as Nintendo^®^ Wii), which allows for little personalization of the activities, and no direct access to raw data. In this sense, the use of an ad-hoc system (e.g., the VRRS) could actually enrich both the exercises proposed and the extraction and analysis of raw data.

To summarize, the impact that postural control has on activities of daily living transforms simple actions, such as chasing a friend or playing in the playground, into real challenges that require enormous physical and cognitive effort, with consequences on daily life [40]. Yet, the possibility to quantitatively assess and train one’s postural control in a motivating context could increase the participation and the self-esteem of children with DCD with their peers. In addition, the improvement of postural control could afford these children the opportunity to refine more advanced and complex skills, such as those required for taking part in sports activities.

This innovative frontier in DCD subjects makes it possible to reduce the gap not only between the abilities but also the possibilities of children with DCD compared with TD children.

Therefore, further studies will be useful to confirm our findings in particular for considering larger sample groups, comorbidities and previous experience with technological tools.

## 5. Conclusions

This study confirms the use of the VRRS stabilometric balance platform as a suitable tool to analyze and assess postural control in children with DCD compared to TD children, as also shown in our previous work [23]. Furthermore, it has been demonstrated that this system is usable in a non-hospital environment, is easily transported, and it collects all the data.

The VRRS also proves to be effective in measuring COP and in discriminating postural control difficulties in children with DCD, correlating significantly with Movement ABC-2 clinical scale scores. This is an important result which would allow the VRRS assessment to be used in conjunction with clinical scales to complete the traditional assessment.

Furthermore, the VRRS system, which is suitable both for assessment and rehabilitation, is also advantageous in that it can be used for tele-rehabilitation. A home-based tailored training programme for postural control could be viable for some children. Finally, once postural control has been evaluated quantitively, there are several solutions for its improvement, which could impact on the quality of life of children with DCD, in terms of children’s functioning, participation in social life, and sharing activities with peers.

## Figures and Tables

**Figure 1 children-09-00957-f001:**
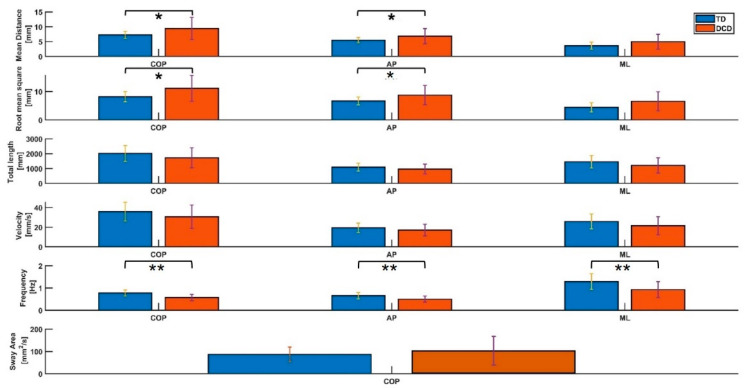
Comparison of the mean values of VRRS parameters and their components between TD and DCD groups. Typically developing (TD) and Developmental Coordination Disorder (DCD). Mean distance (MD), root mean square (RMS), total length (ESC), velocity (VEL), frequency (FREQ), sway area (SWAY). * = *p* < 0.05 and ** = *p* < 0.01.

**Table 1 children-09-00957-t001:** VRRS parameters are reported and divided into their three components, COP, AP, and ML; the Sway area is represented by a single value, related to the COP path.

VRRS Parameters	Parameters Short Name	Centre of Pressure	Anterior Posterior	Medial Lateral
Mean Distance	MD	MD_COP_	MD_AP_	MD_ML_
Root Mean Square of Distance	RMS	RMS_COP_	RMS_AP_	RMS_ML_
Total Excursion	ESC	ESC_COP_	ESC_AP_	ESC_ML_
Velocity	VEL	VEL_COP_	VEL_AP_	VEL_ML_
Frequency	FREQ	FREQ_COP_	FREQ_AP_	FREQ_ML_
Sway area	SWAY	SWAY		

**Table 2 children-09-00957-t002:** Statistical *t*-test results.

VRRS Parameters	Mean Value		Standard Deviation		Standard Error		*p*-Value
	TD	DCD	TD	DCD	TD	DCD	
MD_COP_ [mm]	7.57	10.59	2.29	6.42	0.41	1.47	**0.020 ***
MD_AP_ [mm]	5.49	7.22	1.27	2.95	0.23	0.69	**0.029 ***
MD_ML_ [mm]	3.98	6.01	1.99	5.05	0.36	1.19	0.055
RMS_COP_ [mm]	8.93	12.78	2.89	8.36	0.52	1.97	**0.025 ***
RMS_AP_ [mm]	6.98	9.27	1.64	3.95	0.30	0.93	**0.030 ***
RMS_ML_ [mm]	5.18	8.23	2.82	7.88	0.51	1.85	0.059
ESC_COP_ [mm]	2080.49	2015.76	629.18	1064.45	114.87	250.89	0.817
ESC_AP_ [mm]	1106.26	1124.59	274.43	571.04	50.10	134.59	0.900
ESC_ML_ [mm]	1514.18	1411.92	528.16	784.49	96.42	184.90	0.592
VEL_COP_ [ms^−1^]	36.98	35.86	11.23	18.98	2.05	4.47	0.822
VEL_AP_ [ms^−1^]	19.66	20.00	4.88	10.19	0.89	2.40	0.894
VEL_ML_ [ms^−1^]	26.92	25.12	9.43	13.99	1.72	3.29	0.597
SWAY [mm^2^s^−1^]	105.61	164.35	68.03	198.37	12.42	46.75	0.239
FREQ_COP_ [Hz]	0.81	0.57	0.15	0.13	0.02	0.03	**0.000 ****
FREQ_AP_ [Hz]	0.66	0.50	0.13	0.13	0.02	0.03	**0.000 ****
FREQ_ML_ [Hz]	1.42	0.93	0.49	0.35	0.08	0.08	**0.001 ****

Typically developing (TD) and Developmental Coordination Disorder (DCD). Mean distance (MD), root mean square (RMS), total length (ESC), velocity (VEL), frequency (FREQ), sway area (SWAY). Centre Of Pressure (COP), Anterior-posterior (AP) and Medial-Lateral (ML). ***** = *p* < 0.05; ****** = *p* < 0.01.

**Table 3 children-09-00957-t003:** Pearson correlation results between the VRRS parameters and the MABC-2 scores.

VRRS Parameters	MABC-2 Balance Standard Score		MABC-2 Balance Percentile Score		MABC-2 Total Standard Score		MABC-2 Total Percentile Score	
	ρ	*p*-value	ρ	*p*-value	ρ	*p*-value	ρ	*p*-value
MD_COP_	**−0.428 ****	**0.003**	**−0.359 ***	**0.014**	**−0.399 ****	**0.006**	**−0.305 ***	**0.039**
MD_AP_	**−0.388 ****	**0.008**	**−0.335 ***	**0.023**	**−0.384 ****	**0.008**	**−0.299 ***	**0.044**
MD_ML_	**−0.415 ****	**0.004**	**−0.350 ***	**0.017**	**−0.374 ****	**0.011**	**−0.290 ***	**0.050**
RMS_COP_	**−0.422 ****	**0.004**	**−0.348 ***	**0.018**	**−0.396 ****	**0.006**	**−0.297 ***	**0.045**
RMS_AP_	**−0.384 ****	**0.008**	**−0.337 ***	**0.022**	**−0.381 ****	**0.009**	**−0.298 ***	**0.044**
RMS_ML_	**−0.414 ****	**0.004**	**−0.338 ***	**0.022**	**−0.377 ****	**0.010**	**−0.291 ***	**0.050**
ESC_COP_	−0.020	0.895	−0.005	0.975	0.001	0.997	0.033	0.829
ESC_AP_	−0.020	0.894	−0.032	0.834	−0.027	0.858	−0.018	0.911
ESC_ML_	−0.010	0.950	0.020	0.894	0.027	0.860	0.069	0.651
VEL_COP_	−0.019	0.902	0.004	0.979	0.001	0.993	0.033	0.829
VEL_AP_	−0.019	0.902	−0.031	0.840	−0.026	0.863	−0.017	0.911
VEL_ML_	−0.008	0.955	0.021	0.891	0.027	0.858	0.068	0.651
SWAY	−0.286	0.054	−0.227	0.129	−0.279	0.060	−0.198	0.188
FREQ_COP_	**0.695 ****	**0.000**	**0.673 ****	**0.000**	**0.699 ****	**0.000**	**0.643 ****	**0.000**
FREQ_AP_	**0.533 ****	**0.000**	**0.468 ****	**0.001**	**0.532 ****	**0.000**	**0.439 ****	**0.002**
FREQ_ML_	**0.614 ****	**0.000**	**0.629 ****	**0.000**	**0.604 ****	**0.000**	**0.587 ****	**0.000**

Mean distance (MD), root mean square (RMS), total length (ESC), velocity (VEL), frequency (FREQ), sway area (SWAY). Centre Of Pressure (COP), Anterior-posterior (AP) and Medial-Lateral (ML). Movement Assessment Battery for Children-second edition (MABC-2). Pearson Coefficient (ρ) ***** = *p* < 0.05; ****** = *p* < 0.01.

## Data Availability

The raw data supporting the conclusions of this article will be made available by the authors, without undue reservation.

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
