# Peer review of "New Technological Approach for the Evaluation of Postural Control Abilities in Children with Developmental Coordination Disorder"

_children, 2022, doi:10.3390/children9070957_

Round 1

Reviewer 1 Report

In the reviewed article, the authors present the results of research on postural control skills in children with DCD and typically developing children (TD) using the Virtual Reality Rehabilitation System (VRRS).  Undoubtedly, the subject of this study is very important given the increasing number of children with DCD. Moreover, the use of VRRS offers new possibilities, as this tool can be used outside the clinical settings as part of tele-rehabilitation treatment. From a methodological point of view, the research was conducted in a correct form. Also, presentation of the results does not raise any significant reservations. Nevertheless, in order to increase the scientific value of the article, I propose to introduce the following corrections and supplements to the text:

1. In the following sentence: ‘Therefore, the aim of this study is to investigate and analyze postural control abilities in children with DCD compared to peers using the VRRS system’ (page 2, lines 95-96), authors should identify the peers they mean.

2. Authors should expend the "Limitations" section (page 8) in which it should be emphasized that their research is based on relatively small number of the subjects. Additionally, the age of the respondents, especially in the group of children with DCD, is too varied. For these reasons, the value of the final conclusions is limited.

3. The text contains a certain number of linguistic errors, mainly stylistic, so it should be corrected by a 'native speaker'. Some examples: 1. On page 1 (lines 25-26) instead of ‘A t-test to identify differences in the VRRS parameters between the two groups was performed’. It should be ‘A test-t was performed to identify differences in VRRS parameters between the two groups’. 2. On page 2 (lines 77-78) instead of ‘The most commonly used instrumented tool to quantify the stability in standing position[18]– [20] is posturography; …’, it should be ‘The instrumented tool most commonly used to quantify stability in standing position [18]-[20] is posturography, …’. 3. On page 2 (lines 82-83) instead of ‘Despite this tool being valid for acquiring quantitative data, there is a high heterogeneity among the administered protocols’, it should be ‘Although this tool is valid for the acquisition of quantitative data, there is a high heterogeneity among the protocols administered’. There are many more such examples.

After taking into account my suggestions, I support the publication of this article.

Reviewer 2 Report

It is a very informative study about a type of developmental disability that is less regarded. Developmental Coordination Disorder (DCD), with its high prevalence rate, deserves more attention from professionals in the field of developmental disabilities.

The paper presented a concise definition of DCD and its criteria. I think considering the following points or comments will help the intelligibility of the submitted data.

·        Please present a definition for the “force platform” and why it was a gold standard evaluation.

·        Please clarify the word “peer” in the final part of the introduction and the study’s aim. Particularly in the last sentence of that part, “…to investigate and analyze postural control abilities in children with DCD compared to peers using the VRRS system”. Do you mean typically developing peers or peers with no DCD but possibly with other types of developmental disabilities?

·        I recommend providing more data on recruiting the sample. How did they inform about the study? How many approached and how many accepted to participate?

·        What were the inclusion and exclusion criteria for the DCD and typically developing group to be allocated to each group? It is also imperative to mention the number (percentage) who were possibly excluded because of not meeting the predetermined criteria.

·        It is also important to mention in this part who did evaluate individuals with DCD and how did they evaluate? In a clinical session, indirectly and through parents’ and caregivers’ information or…if different evaluators engaged in this process, what was the protocol for consistencies among different evaluators? (please provide more information regarding what is written in the manuscript as “the same assessment protocol”)

·        It is also very crucial to mention that having previous experience with VRRS system or similar systems for the participants were considered or not. If it is not considered, it needs to be mentioned in the limitation part and suggestions for further studies to understand the impacts of having previous experience in performing the requested tasks.

·        From my point of view, what is mentioned as the limitation of the present study was out of the scope of the study. You were about to evaluate the postural abilities of two groups of individuals, therefore, considering different sensory conditions. This is not a limitation because, as you mentioned, the available data indicates a significant difference between the sensory abilities of children with and without DCD.   You also raised many court-worthy reasons for considering a balance test with open eyes that is satisfactory. Other limitations deserve to be mentioned, for an example, sampling issues with viewing a convenient sample, using typically developing as a general term for the non-DCD group without evaluating for the existence of possible another diagnosis, underrating the factor of having previous experience with similar VRRS systems technologies (i.e., Kinect XBOX or other systems “you mentioned Nintendo” that may have the impact of performance level).

·        I also could not figure out the term “bipodal static,” and “bipodal balance “ and t is very helpful to provide more information about it.

Reviewer 3 Report

It is a pleasure to review this well-crafted manuscript, there are only a few issues to review:

- Line 38: reference DSM V.

- There are word join errors or excess spaces, e.g. in the summary (lines 29 and 31). Review the rest of the manuscript.

- Unify references (line 78).

- Page 6/13 remains blank, review.

On the other hand, I would like to answer some questions:

- Couldn't there be a bias in not taking into account, in both groups, if there was previous experience with new technologies that could influence the results?

- It seems evident that children with typical development, who are also in a roller-skating school, are going to have better scores in the evaluations than children with DCD. Why was the selection of the sample of children with typical development in this school made?

Round 2

Reviewer 2 Report

This is an improved version of the manuscript. The authors have done their best to address the issues with the previously submitted version. The only possible remaining issue is some minor writing style that the editorial might be able to address. As an example, I can mention the line 320 in which it is written:

iii) the sample study of children was composed by a small number of participants with 320 different range age. 

that needs to be changed to:

iii) the sample study of children was composed of a small number of participants of 320 different range ages.